# Bridging verbal coordination and neural dynamics

**Isaïh Schwab-Mohamed[1,2]\*, Manuel R Mercier[1], Agnès Trebuchon[1,3], Benjamin Morillon[1], Leonardo Lancia[4], Daniele Schön[1]\***

[1]Aix Marseille University, Inserm, INS, Inst Neurosci Syst, Marseille, France; [2]Aix-Marseille Univ, Institute of Language, Communication and the Brain, Marseille, France; [3]APHM, Hôpital de la Timone, Service de Neurophysiologie Clinique, Marseille, France; [4]Aix Marseille University, CNRS, Laboratoire Parole et Langage (LPL), Aix-en-Provence, France

**\*For correspondence:**
isaih.schwab.mohamed@outlook.com (IS-M);
daniele.schon@univ-amu.fr (DS)

**Competing interest:** The authors declare that no competing interests exist.

## eLife Assessment

This paper reports on an **important** study that aims to move beyond current experimental approaches in speech production by (1) investigating speech in the context of a fully interactive task and (2) employing advanced methodology to record intracranial brain activity. Together these allow for examination of the unfolding temporal dynamics of brain-behaviour relationships during interactive speech. This approach and the analyses presented in support of the authors' claims pose **convincing** evidence.

**Abstract** Our use of language, which is profoundly social in nature, essentially takes place in interactive contexts and is shaped by precise coordination dynamics that interlocutors must observe. Thus, language interaction is highly demanding on fast adjustment of speech production. Here, we developed a real-time coupled-oscillators virtual partner (VP) that allows – by changing the coupling strength parameters – to modulate the ability to synchronise speech with a virtual speaker. Then, we recorded the intracranial brain activity of 16 patients with drug-resistant epilepsy while they performed a verbal coordination task with the VP. More precisely, patients had to repeat short sentences synchronously with the VP. This synchronous speech task is efficient to highlight both the dorsal and ventral language pathways. Importantly, combining time-resolved verbal coordination and neural activity shows more spatially differentiated patterns and different types of neural sensitivity along the dorsal pathway. More precisely, high-frequency activity (HFa) in left secondary auditory regions is highly sensitive to verbal coordinative dynamics, while primary regions are not. Finally, while bilateral engagement was observed in the HFa of the inferior frontal gyrus BA44 – which seems to index online coordinative adjustments that are continuously required to compensate deviation from synchronisation – interpretation of right hemisphere involvement should be approached cautiously due to relatively sparse electrode coverage. These findings illustrate the possibility and value of using a fully dynamic, adaptive, and interactive language task to gather deeper understanding of the subtending neural dynamics involved in speech perception, production as well as their interaction.

## Introduction

Language has been most frequently studied by separately assessing perception and production in an isolated context, in contrast to the interactive context that characterises best its daily usage. The use of language in an interactive context, particularly during conversations, calls upon numerous predictive

and adaptive processes (*Pickering and Gambi, 2018*). Importantly, analyses of conversational data have highlighted the phenomenon of interactive alignment (*Pickering and Garrod, 2004*), illustrating that during a verbal exchange interlocutors tend to imitate each other and to align their linguistic representations on several levels including phonetic, syntactic, or semantic (*Garrod and Pickering, 2009*). This is an unconscious and dynamic phenomenon that possibly renders exchanges between speakers more fluid (*Marsh et al., 2009*). It consists of mutual anticipation (prediction) and coordination of speech production, leading, for instance, to a reduction of turn-taking durations (*Levinson, 2016*; *Corps et al., 2018*).

Recently, in an effort to assess speech and language in more ecological contexts, researchers in neuroscience have used interactive paradigms to study some of these coordinative phenomena. These studies involved turn-taking behaviours such as alternating naming tasks (*Mukherjee et al., 2019*), questions and answers investigating motor preparations (*Bögels et al., 2015*), or manipulating the turn predictability in end-of-turn detection tasks (*Magyari et al., 2014*; for a review see *Bögels and Levinson, 2017*). However, the synchronous speech paradigm has been overlooked. This paradigm requires the simultaneous and synchronised production of the same word, sentence, or text between two people. Interestingly, this task is remarkably well performed without any particular training, both between a speaker and a recording as well as between two speakers (*Cummins, 2002*; *Cummins, 2003*; *Assaneo et al., 2019*). As in most joint tasks, individuals must mutually adjust their behaviour (here speech production) to optimise coordination (*Cummins, 2009*). Furthermore, synchronous speech favours the emergence of alignment phenomena, such as the fundamental frequency or the syllable onset (*Assaneo et al., 2019*; *Bradshaw and McGettigan, 2021*; *Bradshaw et al., 2023*; *Bradshaw et al., 2024*). Overall, synchronous speech represents a strong interactive framework allowing a good level of experimental control. It offers several possibilities for neurophysiological investigation of both speech perception and production and is an interesting case to consider for models of speech motor control (*Figure 1*).

Synchronous speech resembles to a certain extent delayed/altered auditory feedback tasks, which involve real-time perturbations in the speech production signal (such as changes in fundamental frequency and delay). These tasks can induce speech errors as well as modulations in speech and voice features (*Stuart et al., 2002*; *Yamamoto and Kawabata, 2014*; *Karlin et al., 2021*). Additionally, these tasks provide insights into predictive models of speech motor control, where the brain generates an internal estimate of production and corrects errors when auditory feedback deviates from the estimate (*Hickok et al., 2011*; *Houde and Nagarajan, 2011*; *Tourville and Guenther, 2011*; *Ozker et al., 2022*; *Floegel et al., 2023*). Previous studies have revealed increased responses in the superior temporal regions compared to normal feedback conditions (*Hirano et al., 1997*; *Hashimoto and Sakai, 2003*; *Takaso et al., 2010*; *Ozker et al., 2022*; *Floegel et al., 2020*; see *Meekings and Scott, 2021* for a review of error-monitoring and feedback control in the STG during speech production). However, synchronous speech paradigms allow for the investigation of the neural bases of coordinative behaviour rather than of error correction.

So far, the precise spectro-temporal dynamics and spatial distribution of the cortical networks underlying speech coordination remain unknown. To address this issue, we first developed a real-time coupled-oscillator virtual partner (VP) that allows – by changing the coupling strength parameters – to modulate the ability to synchronise speech with a speaker. The VP (*Lancia et al., 2017*) and the synchronous speech task were first tested on a control group to ensure the ability of the virtual agent to coordinate its speech production in real time with the participants. In certain conditions, the agent was programmed to actively cooperate with the participants by synchronising its syllables with theirs. In other conditions, the agent was programmed to deviate from synchronization by producing its syllables between those of the participants. As a result, participants were constantly required to adapt their verbal productions in order to maintain synchronization. Appropriate tuning of the coupling parameters of the virtual agent enabled us to create a variable context of coordination yielding a broad distribution of phase delays between the speaker and agent productions. Subsequently, we leveraged the excellent spatial sensitivity and temporal resolution of stereotactic depth electrode recordings and acquired neural activity from 16 patients with drug-resistant epilepsy while they performed the adaptive synchronous speech task with the VP.

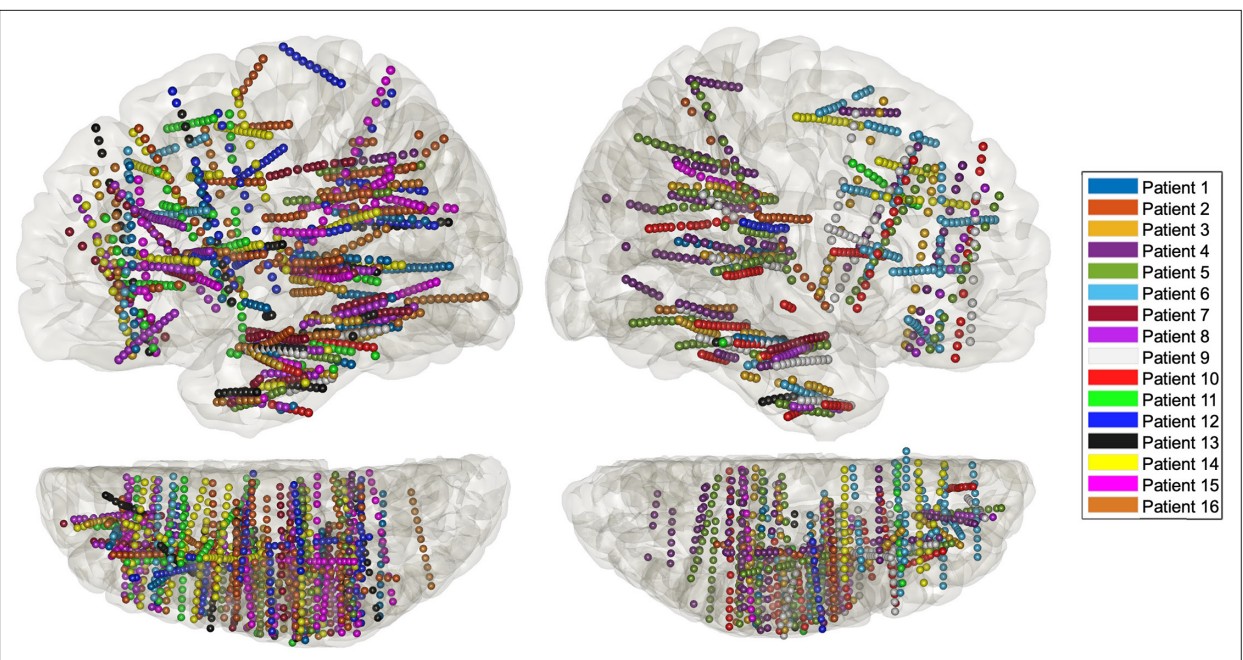

**Figure 1.** Anatomical localisation of the stereotactic EEG (sEEG) electrodes for each patient projected in MNI space on the lateral 3D view (top) and on the top view (*N* =16).

## Results

### Speech coordinative behaviour

The present synchronous speech task allowed us to create a more or less predictable context of coordination, with the objective of obtaining a wide range of coordination variability. Indeed, the intrinsic nature of the task (speech coordination) on one side and the variable coupling parameter of the VP on the other, requires continuous subtle adjustments of the participants' speech production. The degree of coordination between speech signals (VP and participant) was assessed at the syllabic level by computing the phase locking value between the speech temporal envelopes, a measure of the strength of the interaction (or coupling) of two signals. This metric, the verbal coordination index (VCI), is a proxy of the quality of the performance in coordinating speech production with the VP. Overall, the coordinative behaviour was affected, for both controls and patients, by the coupling parameters of the model with a better coordination when the VP was set to synchronise with participants compared to when it was set to speak with a 180° syllabic shift (controls: *t* ratio = 4.55, p ≤ 0.0001; patients: *t* ratio = 6.53, p ≤ 0.0001, see boxplots in *Figure 2C*). This produced, as desired, a rather large coupling variability (controls: range: 0.06–0.84; mean: 0.49; median: 0.49; patients range: 0.11–0.94; mean: 0.55; median: 0.54, see *Figure 2C*). This variability was also present at the individual level (see, for patients only, *Figure 2—figure supplement 1*). Nonetheless, while variable, the coordinative behaviour was significantly better than chance for every patient (see *Figure 2—figure supplement 1*).

### Synchronous speech strongly activates the language network from delta to high gamma range

To investigate the sensitivity of synchronous speech in generating spectrally resolved neural responses, we first analysed the neural responses in both a spatially and spectrally resolved manner with respect to a resting-state baseline condition. In the left hemisphere, neural responses are present in all six canonical frequency bands, from the delta range (1–4 Hz) up to high-frequency activity (HFa, 70–125 Hz, see *Figure 3A*) with medium to large modulation (increase or decrease) in activity compared to baseline (*Figure 3A*). More precisely, while theta, alpha, and beta bands show massive desynchronisation, in particular in the STG BA41/42 (primary auditory cortex), STG BA22 (secondary auditory cortex), and inferior frontal gyrus (IFG) BA44 (Broca's area), the low gamma and HFa bands are dominated by

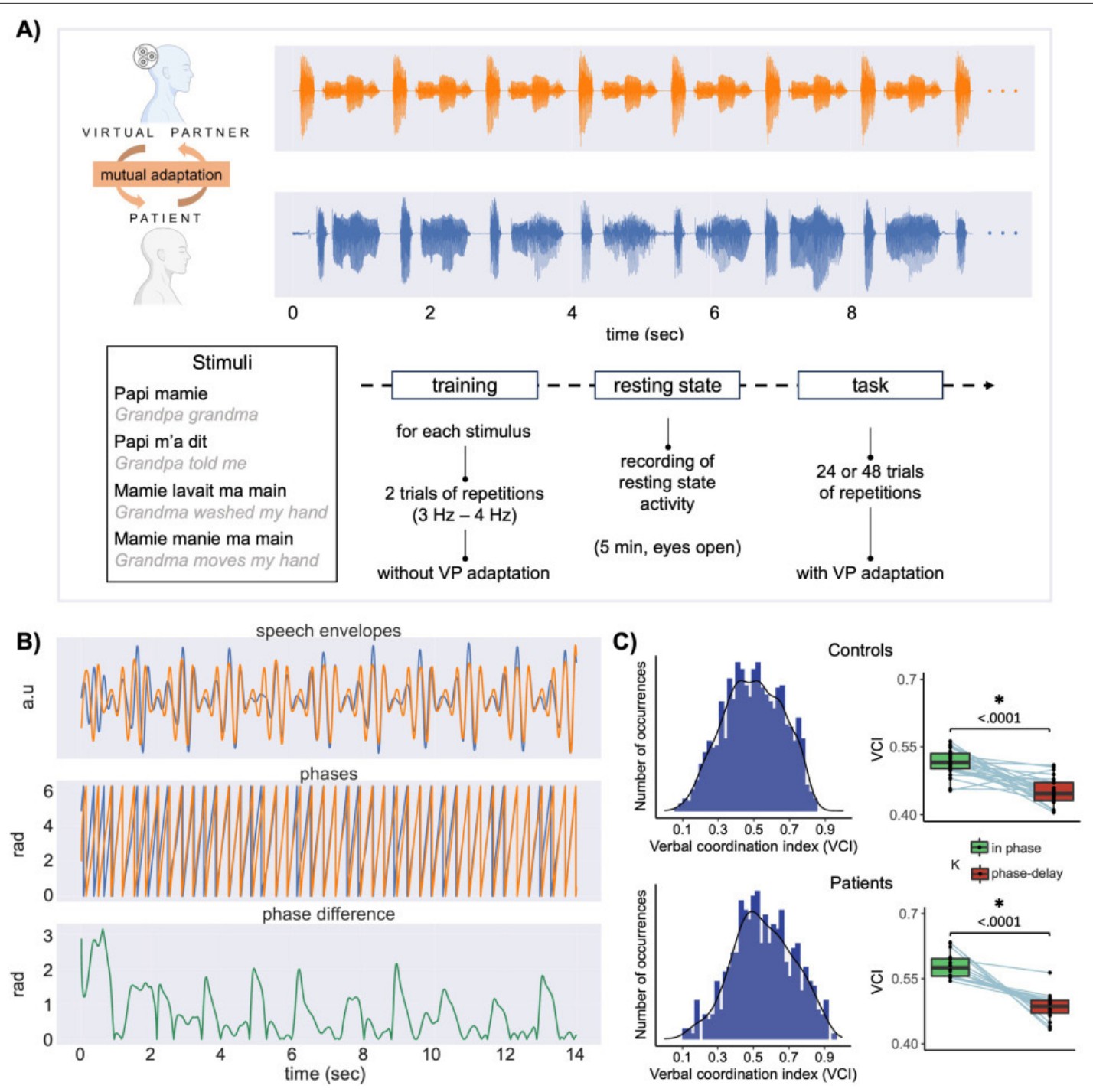

**Figure 2.** Paradigm and coordination indices. (**A**) Top: illustration of one trial of the interactive synchronous speech repetition task (orange: virtual partner [VP] speech; blue: participant speech; stimulus papi m'a dit repeated 10 times; only the 10 first seconds are represented). Bottom: the four speech utterances used in the task and the experimental procedure. (**B**) Speech signals processing stages. The top panel corresponds to the speech envelope, the second to the phase of speech envelope and the third panel to the phase difference between VP and participant speech envelopes, illustrating the coordination dynamics along one trial. (**C**) Left: distributions of verbal coordination index (phase locking values between VP and participant speech envelopes, for each trial) for all participants (top) and patients. Right: boxplots for control participants (top) and patients showing the trial-averaged verbal coordination index as a function of the VP parameters (in-phase coupling vs coupling with a 180° shift).

The online version of this article includes the following figure supplement(s) for figure 2:

**Figure supplement 1.** Distribution of verbal coordination index for each patient (PLV between patient's speech and virtual partner [VP] speech).

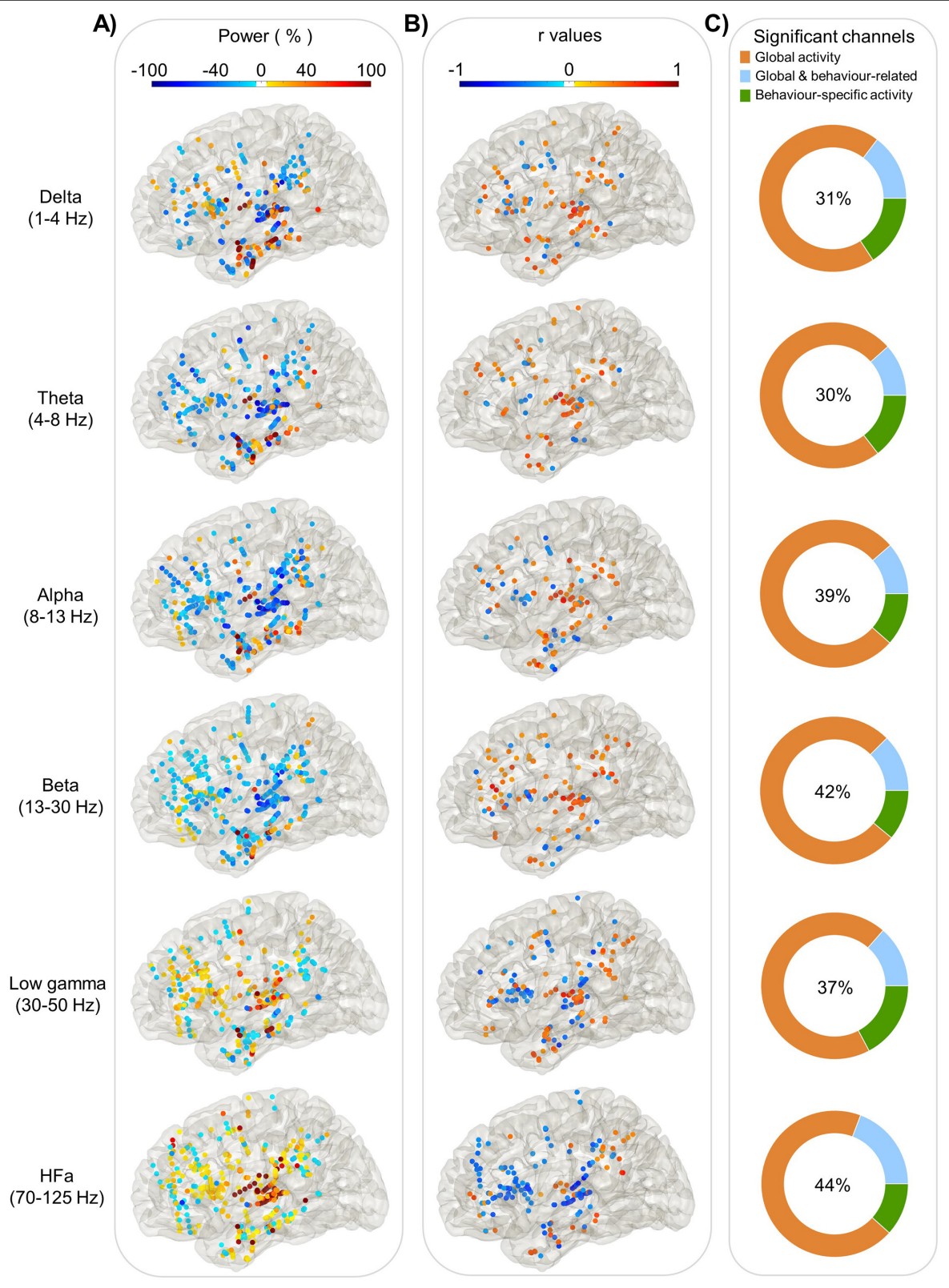

**Figure 3.** Power spectrum analyses and correlation with verbal coordination index (VCI; left hemisphere). Each dot represents a channel where a significant effect was found either on (**A**) Global activity (Task vs Rest) for each frequency band. The activity is expressed in % of power change compared to resting; or on (**B**) Behaviour-related activity: *r* values of the Spearman correlation across trials between the iEEG power and the VCI. (**C**) The proportion of channels where a significant effect was found: in the task versus rest (orange), in the brain–behaviour correlation (green) or for both

*Figure 3 continued on next page*

*Figure 3 continued*

comparisons (blue). The percentage in the centre indicates the overall proportion of significant channels from the three categories with respect to the total number of channels.

The online version of this article includes the following figure supplement(s) for figure 3:

**Figure supplement 1.** Power spectrum analyses and correlation with verbal coordination index (VCI; right hemisphere).

**Figure supplement 2.** Cluster analysis (silhouette score).

power increase in particular in the auditory cortex (STG BA41/42) and in the IFG BA44. This modulation between synchronisation and desynchronisation across frequencies was significant ($F(5) = 6.42$, $p < 0.001$; estimated with linear model using the R function *lm*).

As expected, the whole left hemisphere language network is strongly involved, including both dorsal and ventral pathways (*Figure 3A*). More precisely, in the temporal lobe the superior, middle, and inferior temporal gyri, in the parietal lobe the inferior parietal lobule (IPL) and in the frontal lobe the IFG and the middle frontal gyrus. Similar results are observed in the right hemisphere, neural responses are present across all six frequency bands with medium to large modulation in activity compared to baseline (*Figure 3—figure supplement 1A*) in the same regions. Desynchronisations are present in the theta, alpha, and beta bands while the low gamma and HFa bands show power increases.

## The brain needs behaviour: global versus behaviour-specific neural activity

Comparing the overall activity during the task to activity during rest gives a broad view of the network involved in verbal coordination. However, as stated above, the task was conceived to engender a rather wide range of verbal coordination across trials for each participant. This variety of coordinative behaviours allows to explore the link between verbal and simultaneous neural activity, by computing the correlation across trials between the VCI and the mean power (see methods). This analysis allows us to estimate the extent to which neural activity in each frequency band is modulated as a function of the quality of verbal coordination.

*Figure 3B* shows the significant *r* values (Spearman correlation) for each frequency band in the left hemisphere (see *Figure 3—figure supplement 1B* for the right hemisphere). The first observation is a gradual transition in the direction of correlations as we move up frequency bands, from positive correlations at low frequencies to negative ones at high frequencies ($F(5) = 2.68$, $p = 0.02$). This effect, present in both hemispheres, mimics the reversed desynchronisation/synchronisation process in low- and high-frequency bands reported above. In other words, while in the low-frequency bands stronger desynchronisation goes along with weaker verbal coordination, in the HFa stronger activity is associated with weaker verbal coordination.

Importantly, compared to the global activity (task vs rest, *Figure 3A*), the neural spatial profile of the behaviour-related activity (*Figure 3B*) is more clustered, in the left hemisphere. Indeed, silhouette scores are systematically higher for behaviour-related activity compared to global activity, indicating greater clustering consistency across frequency bands ($t(106) = 7.79$, $p < 0.001$, see *Figure 3—figure supplement 2*). Moreover, silhouette scores are maximal, in particular for HFa, for five clusters ($p < 0.001$), located in the IFG BA44, the IPL BA40 and the STG BA41/42 and BA22 (see *Figure 3—figure supplement 2*).

Comparing in both hemispheres global activity and behaviour-related activities shows that, for each frequency band, approximately 2/3 of the channels are only significant in the global activity (*Figure 3C*, orange part). Of the remaining third, half of the significant channels show a modulation of power compared to baseline that also significantly correlates with the quality of behavioural synchronisation (*Figure 3C*, blue part). The other half are only visible in the brain–behaviour correlation analysis (*Figure 3C*, green part, behaviour-specific).

## Spectral profiles in the language network are nuanced by behaviour

In order to further explore the brain–behaviour relation in an anatomically language-relevant network, we focused on HFa and on those regions (ROI) of the dorsal pathway recorded in at least seven patients for the left hemisphere and at least five patients for the right hemisphere: STG BA41/42

(primary auditory cortex), STG BA22 (secondary auditory cortex), IPL BA40 (inferior parietal gyrus), and IFG BA44 (inferior frontal gyrus, Broca's region). To balance data completeness and statistical power, we included only brain regions recorded in at least seven patients (~44% of the cohort) for the left hemisphere and at least five patients for the right hemisphere (~31% of the cohort), ensuring sufficient representation while minimizing biases due to sparse data. Within each ROI, using Spearman correlation, we quantified the link neural activity and the degree of behavioural coordination. *Figure 4B* (left hemisphere) shows a dramatic decrease of HFa along the dorsal pathway. While left-STG BA41/42 presents the strongest power increase (compared to baseline), it shows no significant correlation with verbal coordination ($t(28) = -1.81$, $p = 0.08$; Student's $t$-test, FDR correction). By contrast, the left-STG BA22 shows both a significant power increase in the HFa and a significant negative correlation between HFa and behaviour (i.e. VCI) ($t(29) = -4.40$, $p < 0.0001$; Student's $t$-test, FDR correction), marking a fine distinction between primary and secondary auditory cortex. Finally, the brain–behaviour correlation is maximal in the left-IFG BA44 ($t(26) = -5.60$, $p < 0.0001$; Student's $t$-test, FDR correction).

The decrease in HFa along the dorsal pathway is replicated in the right hemisphere (*Figure 4— figure supplement 1*). However, while both the right STG BA41/42 and STG BA22 present a power increase (compared to baseline) – with a stronger increase for the STG BA41/42 – neither shows a significant correlation with verbal coordination ($t(45) = -1.65$, $p = 0.1$; $t(8) = -0.67$, $p = 0.5$; Student's $t$-test, FDR correction). By contrast, results in the right IFG BA44 are similar to the one observed in the left hemisphere with a significant power increase associated with a negative brain–behaviour correlation ($t(17) = -3.11$, $p = 0.01$; Student's $t$-test, FDR correction).

## The IFG is sensitive to speech coordination dynamics

To model the temporal dynamics of the relation between verbal coordination and neural activity, we conducted a behaviour–brain phase-amplitude coupling (PAC) analysis at the single trial level. That is, we used the power of high-frequency neural activity (HFa: 70–125 Hz) and the low-frequency phase of the behaviour; the latter being either the phase of the speech signals of the VP or the verbal coordination dynamics (i.e. the phase difference between VP and speaker).

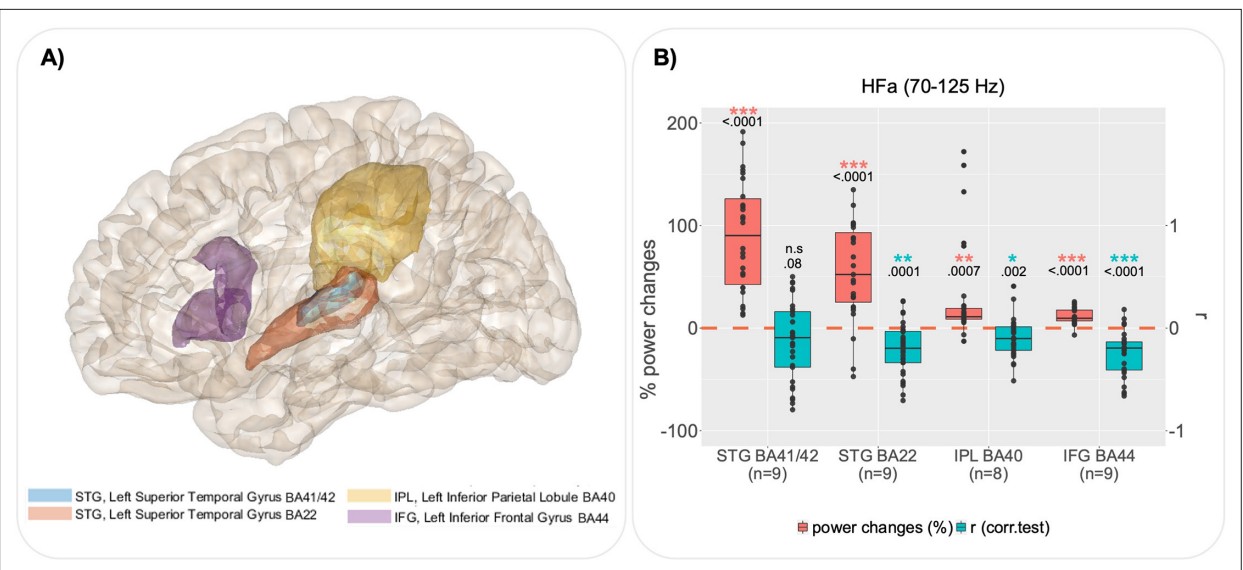

**Figure 4.** Group analysis by regions of interest (ROI) for the left hemisphere. (**A**) ROI defined according to the cluster analysis (see *Figure 3—figure supplement 2*), the delimitation of regions is based on the Brainnetome atlas. (**B**) For each ROI, boxplots illustrate, in red, channels with significant global power changes (high-frequency activity [HFa], task vs rest) and, in blue, their corresponding r values (correlation between HFa power and verbal coordination index, VCI). Red and blue stars indicate a significant difference from a null distribution. Dots represent independent iEEG channels. The 'n' below each ROI specifies the number of patients. STG: superior temporal gyrus; IPL: inferior parietal lobule; IFG: inferior frontal gyrus; BA: Brodmann area.

The online version of this article includes the following figure supplement(s) for figure 4:

**Figure supplement 1.** Group analysis by regions of interest (ROI; right hemisphere).

When looking at the analysis of the left hemisphere (*Figure 5A*), coupling is strongest as expected in the auditory regions (STG BA41/42 and STG BA22), but it is also present in the left IPL and IFG. Notably, when comparing – within the ROI previously described – the PAC with the VP speech and the PAC with the phase difference, the coupling relationship changes when moving along the dorsal pathway: a stronger coupling in the auditory regions with the speech input, no difference between speech and coordination dynamics in the IPL and a stronger coupling for the coordinative dynamics compared to speech signal in the IFG (*Figure 5B*). When looking at the right hemisphere, we observe the same changes in the coupling relationship when moving along the dorsal pathway, except that no difference between speech and coordination dynamics is present in the right secondary auditory regions (STG BA22; *Figure 5—figure supplement 1*). Similar results were obtained when using the phase of the patient speech rather than the VP speech (as a control analysis). Finally, in order to assess whether the phase-amplitude relationship is different for anticipatory (negative delays) and compensatory (positive delays) behaviour between the VO and the patients' speech, we assessed the difference between PAC in trials with negative and positive delays (*Figure 5—figure supplement 2*). Although there seems to be a trend in the left IFG with anticipatory behaviour (negative lags) being associated with stronger neural coupling, this difference did not reach significance.

## Discussion

In this study, we investigated speech coordinative adjustments using a novel interactive synchronous speech task (*Lancia et al., 2017*). To assess the relation between speech coordination and neural dynamics, we capitalised on the excellent spatiotemporal sensitivity of human stereotactic recordings (sEEG) from 16 patients with drug-resistant epilepsy while they produced short sentences along with a VP. Critically, the VP was able to coordinate and adapt its verbal production in real time with those of the participants, thus enabling the creation of a variable context of coordination yielding a broad distribution of phase delays between participant and VP productions. Several interesting findings can be emphasised. Firstly, the task involving both speech production and perception is efficient to highlight both the dorsal and ventral pathways, from low-frequency activity to HFa (*Figure 3*). Secondly, spectral profiles of neural responses in the language network are nuanced when combined with behavioural data, highlighting the fact that some regions are involved in the task in a general manner, while others are sensitive to the quality of verbal coordination (*Figure 3*). Thirdly, HFa in left secondary auditory regions shows a stronger sensitivity to behaviour (coordination success) compared to left primary auditory regions (*Figure 4*). Finally, the HFa of the IFG BA44 (bilaterally) specifically indexes the online coordinative adjustments that are continuously required to compensate for deviations from synchronisation (*Figure 5*).

### Left secondary auditory regions are more sensitive to coordinative behaviour

In the left hemisphere, the increase of HFa (as compared to rest) in both the STG BA41/42 (primary auditory cortex) and STG BA22 (secondary auditory cortex) is associated with different cognitive functions. Indeed, when considering whether HFa correlates with the behavioural coordination index – here the phase locking value between the patient and speaker speech – only STG BA22 shows a significant (inverse) correlation with a reduced HFa in strongly coordinated trials. This spatial distinction made between primary and secondary auditory regions in terms of their sensitivity to task demands has already been observed in the auditory cortex, particularly in the HFa (*Nourski, 2017*). The use of auditory target detection paradigms – phonemic categorisation (*Chang et al., 2011*), tone detection (*Steinschneider et al., 2014*), and semantic categorisation tasks (*Nourski et al., 2015*) – where participants are asked to press a button when they hear the target, revealed task-dependent modulations only in the STG (posterolateral superior temporal gyrus) and not in Heschl's gyrus. In these studies, modulations corresponded to increases in activity in response to targets compared to non-targets, which have been interpreted in terms of selective attention or a bias towards behaviourally relevant stimuli (*Petkov et al., 2004*; *Mesgarani and Chang, 2012*). Here, we extend these findings to a more complex and interactive context.

The observed negative correlation between verbal coordination and HFa in left-STG BA22 suggests a suppression of neural responses as the degree of behavioural synchrony increases. This result is

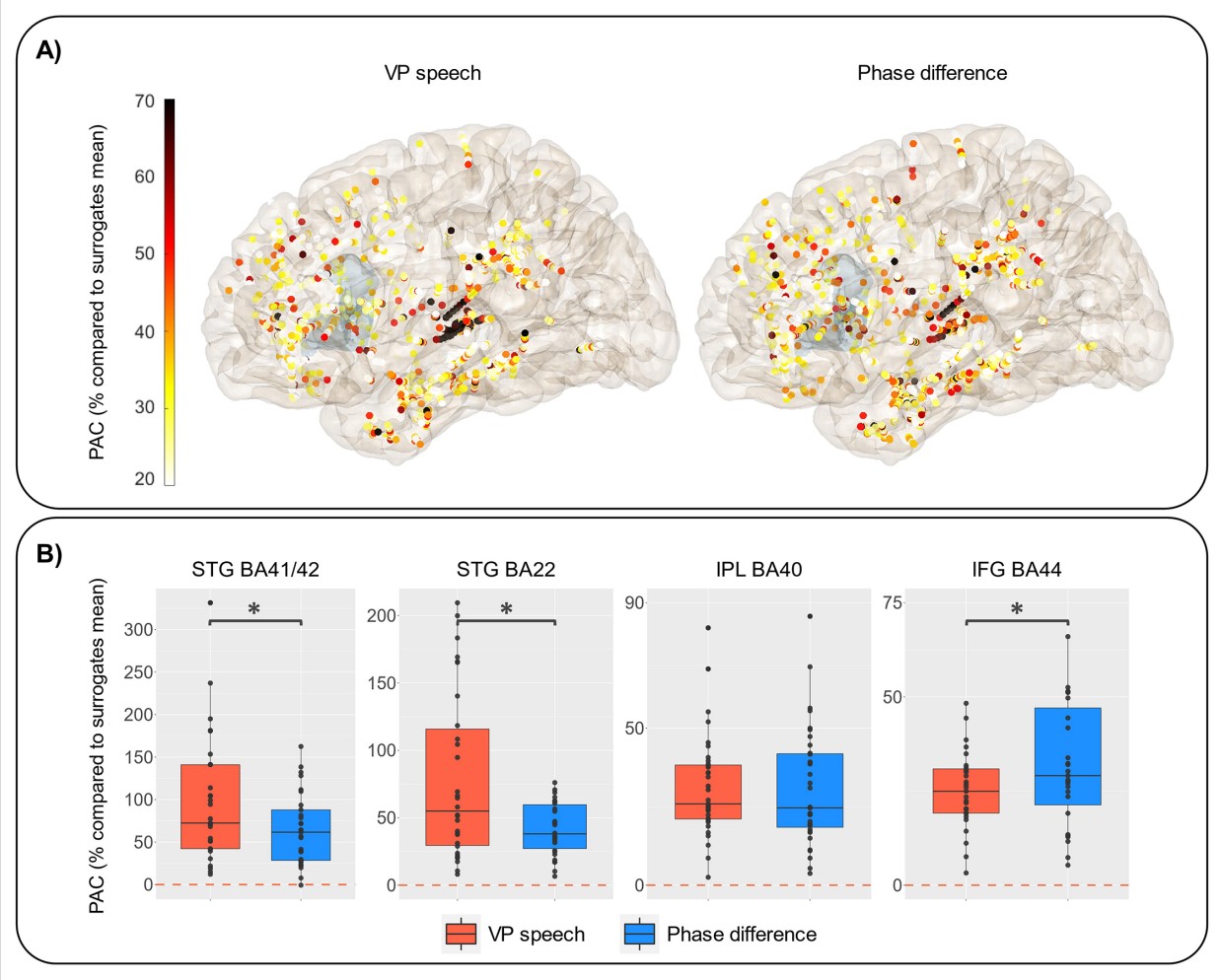

**Figure 5.** Phase-amplitude coupling (PAC) between virtual partner [VP] speech signal or coordination dynamics and high-frequency activity (HFa). (**A**) Representation of the increase in PAC expressed in % compared to surrogates mean when using the VP speech (left) or the coordination dynamics (phase difference between VP and patient, right). The shaded (slight blue) area corresponds to the location of the inferior frontal gyrus (IFG) BA44. (**B**) PAC values for VP (in red) and phase difference (in blue) by regions of interest. Statistical difference between the two types of PAC is calculated using paired Wilcoxon's test (STG BA41/42: p = 0.01; STG BA22: p = 0.004; inferior parietal lobule [IPL] BA40: p = 0.6; IFG BA44: p = 0.02). Y-axis range has been adjusted to better illustrate the contrast between VP speech and coordination dynamics.

The online version of this article includes the following figure supplement(s) for figure 5:

**Figure supplement 1.** Phase-amplitude coupling (PAC) analyses by region of interest (right hemisphere).

**Figure supplement 2.** Phase-amplitude coupling (PAC) according to the behavioural delay (left hemisphere).

reminiscent of findings on speaker-induced suppression (SIS), where neural activity in auditory cortex decreases during self-generated speech compared to externally generated speech (*Meekings and Scott, 2021*; *Niziolek et al., 2013*). However, our paradigm differs from traditional SIS studies in two critical ways: (1) the speaker's own voice is always present and predictable from the forward model, and (2) no passive listening condition was included. Therefore, our findings cannot be directly equated with the original SIS effect. Instead, we propose that the suppression observed here reflects an SIS-related phenomenon specific to the synchronous speech context. Synchronous speech requires simultaneous monitoring of self- and externally generated speech, a task that is both attentionally demanding and coordinative. This aligns with evidence from *Ozker et al., 2022*; *Ozker et al., 2024*, showing that the same neural populations in STG exhibit SIS and heightened responses to feedback perturbations. These findings suggest that SIS and speech monitoring are related processes, where suppressing responses to self-generated speech facilitates error detection. In our study, suppression of HFa as coordination increases may reflect reduced prediction errors due to closer alignment

between perceived and produced speech signals. Conversely, increased HFa during poor coordination may signify greater mismatch, consistent with prediction error theories (*Houde and Nagarajan, 2011*; *Friston et al., 2020*). Furthermore, when self- and externally generated speech signals are temporally and phonetically congruent, participants may perceive external speech as their own. This echoes the 'rubber voice' effect, where external speech resembling self-produced feedback is perceived as self-generated (*Zheng et al., 2011*; *Lind et al., 2014*; *Franken et al., 2021*). While this interpretation remains speculative, future studies could incorporate subjective reports to investigate this phenomenon in more detail.

Furthermore, the absence of correlation in the right STG BA22 (*Figure 4—figure supplement 1*) seems in the first instance to challenge influential speech production models (e.g. *Guenther and Hickok, 2016*) propose that the right hemisphere is involved in feedback control. However, one needs to consider the task at stake heavily relied upon temporal mismatches and adjustments. In this context, the left-lateralised sensitivity to verbal coordination reminds one of the works of Floegel and colleagues (*Floegel et al., 2020*; *Floegel et al., 2023*) suggesting that both hemispheres are involved depending on the type of error: the right auditory association cortex monitoring preferentially spectral speech features and the left auditory association cortex monitoring preferentially temporal speech features. Nonetheless, the right temporal lobe seems to be sensitive to speech coordinative behaviour, confirming previous findings using fMRI (*Jasmin et al., 2016*) and thus showing that the right hemisphere has an important role to play in this type of tasks (e.g. *Jasmin et al., 2016*).

## IFG (BA44) as a site of speech coordination/planning in dynamic context

Our results highlight the involvement of the IFG bilaterally, in particular the BA44 region, in speech coordination. Firstly, trials with a weak verbal coordination (VCI) are accompanied by more prominent HFa (*Figure 4*; *Figure 4—figure supplement 1*). Secondly, when considering the within-trial time-resolved dynamics, the PAC reveals a tight relation between the low-frequency behavioural dynamics (phase) and the modulation of high-frequency neural activity (amplitude, *Figure 5B*; *Figure 5—figure supplement 1*). This relation is strongest when considering the phase adjustments rather than the phase of speech of the VP per se: larger deviations in verbal coordination are accompanied by an increase in HFa. Additionally, we also tested for potential effects of different asynchronies (i.e. temporal delay) between the participant's speech and that of the VP but found no significant differences (*Figure 5—figure supplement 2*). While the lack of delay effect does not permit one to conclude about the sensitivity of BA44 to absolute timing of the partner's speech, its neural dynamics are linked to the ongoing process of resolving phase deviations and maintaining synchrony.

These findings are in line with the importance of higher-level frontal mechanisms for behavioural flexibility and their role in the hierarchical generative models underlying speech perception and production (*Cope et al., 2017*). More precisely, they are in line with works redefining the role of Broca's area (BA44 and BA45) in speech production associating it more to speech planning rather than articulation per se (*Flinker et al., 2015*; *Basilakos et al., 2018*). Indeed, electrodes covering Broca's area show the greatest activity before the onset of articulation and not during speech production. This has been interpreted in favour of a role at a pre-articulatory stage rather than an 'online' coordination of the speech articulators at least in picture naming (*Schuhmann et al., 2009*), word repetition (*Flinker et al., 2015*; *Ferpozzi et al., 2018*), and turn-taking (*Castellucci et al., 2022*). According to these studies, Broca's area may be a 'functional gate' at a pre-articulatory stage, allowing the phonetic translation before speech articulation (*Ferpozzi et al., 2018*). Our use of a synchronous speech task allows to refine this view by showing that these pre-articulatory commands are of continuous rather than discrete nature.

In other terms, the discrete (on–off) and ignition-like behaviour of neuronal populations in Broca's area gating pre-articulatory commands before speech may be due to the discrete nature of the tasks used to assess speech production. Notably, picture naming, word repetition, word reading and even turn-onsets imply that speech production is preceded by a silent period during which the speaker listens to speech or watches pictures. By contrast, the synchronous speech task requires continuous temporal adjustments of verbal productions in order to reach synchronisation with the VP. Relatedly, the involvement of IFG in accurate speech timing has been previously shown via thermal manipulation (*Long et al., 2016*).

Of note, temporal adjustments (prediction error corrections) are also needed for fluent speech in general, beyond synchronous speech, and give rise to the rhythmic nature of speech. Temporal adjustments possibly take advantage of the auditory input and are referred to as audio-motor interactions that can be modelled as a coupled oscillator (*Poeppel and Assaneo, 2020*). Interestingly, during speech perception, the coupling of theta and gamma bands in the auditory cortex reflects tracking of slow speech fluctuations to spiking gamma (*Morillon et al., 2010*; *Morillon et al., 2012*; *Hyafil et al., 2015*; *Lizarazu et al., 2019*; *Oganian and Chang, 2019*; *Leonard et al., 2024*), similar to what we describe in the auditory cortex. By contrast, in the IFG, the coupling in the HFa is strongest with the input–output phase difference (input of the VP − output of the speaker), a metric that could possibly reflect the amount of error in the internal computation to reach optimal coordination. This indicates that this region could have an implication in the optimisation of the predictive and coordinative behaviour required by the task. This well fits with the anatomical connectivity that has been described between Broca's and Wernicke's territory via the long segment of the arcuate fasciculus, possibly setting the base for a mapping from speech representations in Wernicke's area to the predictive proprioceptive adjustments processed in Broca's area (*Catani and ffytche, 2005*; *Oestreich et al., 2018*). This also aligns with broader theories on the relationship between perception and action, such as predictive coding and active inference, which propose shared sensory prediction mechanism and neural computational architecture for both processes (*Friston et al., 2017*; *Friston et al., 2020*).

Finally, while the case of synchronous speech may seem quite far away from real-life conversational contexts, the models describing language interaction consider that listeners covertly imitate the speaker's speech and timely construct a representation of the underlying communicative intention which allows early fluent turn-taking (*Pickering and Gambi, 2018*; *Levinson, 2016*). Moreover, synchronous speech has recently gained interest in the neuroscience field due to important results showing a relation between anatomo-functional features and synchronization abilities. More precisely, *Assaneo et al., 2019* used a spontaneous speech synchronisation (SSS) test wherein participants produce the syllable /tah/ while listening to a random syllable sequence of predefined pace. The authors identified two groups of participants (high and low synchronisers) characterised by their ability to naturally synchronise their productions more or less easily with the auditory stimulus. Importantly, the ability to synchronise correlates to the degree of lateralisation of the arcuate fasciculus, that connects the IFG and the auditory temporal regions, high synchronisers showing greater lateralisation to the left than the low synchronisers. The more this structural connectivity of the arcuate fasciculus is lateralised in the left hemisphere, the more the activity of the IFG is synchronised with the envelope of the audio stimulus of the SSS test, during a passive listening task. A major limitation of EEG and MEG studies is that they are very sensitive to speech production artefacts, which is not the case for iEEG. Thus, the full dynamics of speech interaction are difficult to assess with surface recordings. Our findings extend these results in several manners. Firstly, speech production is not limited to a single syllable, but to complex utterances. Secondly, the input auditory stimulus is not preset but adapts and changes behaviour in real-time as a function of the dynamic of the 'dyad', here patient–VP. Thirdly, and most importantly, iEEG recording allowing speech artefact-free data, we could extend the relation between coordination abilities and anatomical circuitry of the IFG to the neural dynamics of this same region, showing that it plays an important role in the temporal adjustment of speech that is necessary to coordinate to external speech.

To conclude, the present study illustrates the possibility and interest of using a fully dynamic, adaptive, and interactive language task to gather deeper understanding of the subtending neural dynamics involved in speech perception, production as well as their interaction. It is worth noting that the influence of specific speech units, such as consonants versus vowels, on speech coordination remains to be explored. In non-interactive contexts, participants show greater sensitivity during the production of stressed vowels, possibly reflecting heightened attentional or motor adjustments (*Oschkinat and Hoole, 2022*; *Li and Lancia, 2024*). In this study, the VP's adaptation relies on sensitivity to spectral cues, particularly phonetic transitions, with some (e.g. formant transitions) being more salient than others. However, how these effects manifest in an interactive setting remains an open question, as both interlocutors continuously adjust their speech in real time. Future studies could investigate whether coordination signals, such as phase resets, preferentially align with specific parts of the syllable.

# Materials and methods

## Control – participants

Thirty participants (17 women, mean age 24.7 years, range 19–42 years) took part in the study. All were French native speakers with normal hearing and no neurological disorders. Participants provided written informed consent prior to the experimental session and the experimental protocol was approved by the Institutional Review Board of the French Institute of Health (IRB00003888). Five participants (two women) were excluded from analysis for poor signal-to-noise ratio in speech recordings.

## Patients – participants

Sixteen patients (seven women, mean age 29.8 years, range 17–50 years) with pharmacoresistant epilepsy took part in the study. They were included if their implantation map covered at least partially the Heschl's gyrus and had sufficiently intact diction to support relatively sustained language production. All patients were French native speakers. Neuropsychological assessments carried out before stereotactic EEG (sEEG) recordings indicated that they had intact language functions and met the criteria for normal hearing. In none of them were the auditory areas part of their epileptogenic zone as identified by experienced epileptologists. Recordings took place at the Hôpital de La Timone (Marseille, France). Patients provided written informed consent prior to the experimental session and the experimental protocol was approved by the Institutional Review Board of the French Institute of Health (IRB00003888).

## Data acquisition

The speech signal was recorded using a microphone (RODE NT1) adjusted on a stand so that it was positioned in front of the participant's mouth. Etymotic insert earphones (Etymotic Research E-A-R-TONE gold) fitted with 10 mm foam eartips were used for sound presentation. The parameters and sound adjustment were set using an external low-latency (~5 ms) sound card (RME Babyface Pro Fs, *Kim et al., 2020*), allowing a tailored and temporally precise configuration for each participant. A calibration was made to find a comfortable volume and an optimal balance for both the sound of the participant's own voice, which was fed back through the headphones, and the sound of the stimuli. The aim of this procedure was that the patient would subjectively perceive their voice and the VP voice in equal measure. VP voice was delivered at approximately 70 dB.

The sEEG signal was recorded using depth electrodes shafts with a 0.8-mm diameter containing 5–18 electrode contacts (Dixi Medical or Alcis, Besançon, France). The contacts were 2 mm long and were spaced from each other by 1.5 mm. The placement of the electrode implantations was determined solely on clinical grounds. Sixteen patients with a total of 236 electrodes (145 in the left hemisphere) and 2395 contacts (1459 in the left hemisphere, see *Figure 1*). While this gives a rather sparse coverage of the right hemisphere, we decided, due to the rarity of this type of data, to report results for both hemispheres, with figures for the left hemisphere in the main text and figures for the right hemisphere in the supplementary section. All patients were recorded in a sound-proof Faraday cage using a 256-channel amplifier (Brain Products), sampled at 1 kHz and high-pass filtered at 0.16 Hz.

## Stimuli

Four stimuli corresponding to four short sentences were pre-recorded by both a female and a male speaker. This allowed to adapt to the natural gender differences in fundamental frequency (i.e. so that the VP gender matched that of the patients). All stimuli were normalised in amplitude.

Stimuli consisted of four sentences: 'papi mamie' (/*papi mami*/, grandpa grandma)/'papi m'a dit' (/*papi ma di*/, grandpa told me)/'mamie lavait ma main' (/*mami lavɛ ma mɛ̃*/, grandma washed my hand)/'mamie manie ma main' (/*mami mani ma mɛ̃*/, grandma handles my hand). The four sentences purposely differed in terms of number of syllables (4-4-6-6). Moreover, two of them contained deviations from an otherwise repeating phonological pattern: both in /*papi ma di*/ and in /*mami mani ma mɛ̃*/, the repeated opening/closing of the lips is substituted by the formation and release of a tongue constriction. These manipulations made the sentences more or less easy to articulate (easy-medium-easy-medium). Sentence durations were 2.07, 2.11, 3.16, and 2.87 s, respectively, with a syllable rate of ~3 Hz.

## Experimental design

Participants, comfortably seated in a medical chair, were instructed that they would perform a real-time interactive synchronous speech task with an artificial agent (virtual partner, henceforth VP, see next section) that can modulate and adapt to the participant's speech in real time.

The experiment required three steps (*Figure 2A*). Firstly, the sentences were presented in written form and the experimenter verified that each participant could pronounce them correctly. Secondly, a training phase took place. This required repeating each stimulus over and over together with the VP for ~14 s. More precisely, the sentence was first presented on a screen. When the participant pressed the 'space' bar, the visual stimulus went off, and the VP started to 'speak'. The participant was instructed to repeat the stimuli as synchronously as possible with the VP for the whole trial duration. In the training phase, the VP did not adapt to the participant.

The training allowed participants to familiarise with the synchronous speech task but also to build a personalised VP model incorporating the articulatory variability of each participant (see section below). The training was followed by a resting phase, allowing the recording of resting state activity for a period of 5 min. This time also allowed to build, for each participant, the VP model.

The third step was the actual experiment. This was identical to the training but consisted of 24 trials (14 s long, speech rate ~3 Hz, yielding ~1000 syllables). Importantly, the VP varied its coupling behaviour to the participant. More precisely, for a third of the trials, the VP had a neutral behaviour (close to zero coupling: $k = \pm0.01$). For a third, it had a moderate coupling, meaning that the VP synchronised more to the participant speech ($k = -0.09$). And for the last third of the trials, the VP had a moderate coupling but with a phase shift of $\pi/2$, meaning that it moderately aimed to speak in between the participant syllables ($k = +0.09$). The coupling values were empirically determined on the basis of a pilot experiment in order to induce more or less synchronisation but keeping the phase-shifted coupling at a rather implicit level. In other terms, while participants knew that the VP would adapt, they did not necessarily know in which direction the coupling went. Depending on patient fatigue, a second experimental session with 24 extra trials was proposed (for 6 of the 16 patients). The control group participant runs a single session of 48 trials.

## Virtual partner (principles)

The VP used for the experiment allows generation of speech (words or short utterances) while adapting it in real-time to the concurrent speech input. The VP environment is built upon the Psychtoolbox-3 programme and operates within MATLAB, leveraging custom C subroutines to boost its performance. Its operation revolves around a loop whose iterations are executed at consistent intervals of $\Delta t = 4$ ms. During each iteration, the programme analyses the latest segment (25 ms) of speech produced by the participant and streams a portion of speech to the output device. More precisely, at each iteration of the main loop, the functioning of the VP can be described in four steps:

1. A feature vector is extracted from the last chunk of the input signal.
2. The phase value of the input signal chunk is calculated by mapping the input feature vector onto the corresponding vector of the stimulus signal and retrieving the associated phase value. This step is performed using a dynamic-time-warping algorithm (*Dixon, 2005*). To enhance precision, the input chunk is mapped onto several model utterances (all time-aligned with the signal used as a stimulus) that are tailored to the characteristics of the participant's speech in the training phase.
3. A chunk of stimulus signal is chosen to be sent to the output device. The selection is guided by applying the *Kuramoto, 1975* equation to the difference between the phase values representing the current positions of the participant and the VP in their syllabic cycles. This enables real-time lengthening or shortening of speech chunks as needed to adjust towards the preferred phase (0 or $\pi/2$). Significantly, in applying this equation, we assign a specific value to the coupling strength parameter '$k$', linking the behaviour of the VP to that of the participant. Values can vary from trial to trial and could be close to 0 ($k = \pm0.01$) resulting in a neutral behaviour, negative values ($k = -0.09$) equivalent to a moderate coupling behaviour (tendency to synchronise), or positive values ($k = +0.09$) corresponding to a moderate coupling behaviour with a phase shift of $\pi/2$ (tendency to speak in between participant syllables).
4. The selected chunk of stimulus signal is integrated into the output stream via WSOLA synthesis (Waveform Similarity Overlap-Add).

Of note, the coupling strengths were chosen to be rather weak and thus do not allow reaching 0 or $\pi/2$ phase synchrony, but rather yield the desired large panel of phase delays in the VP–participant coordinative behaviour (see *Figure 1*).

## Data analysis

### Speech signal

Speech signals of the participants and the VP were processed using Praat and Python scripts. Firstly, raw speech signals were downsampled from 48 to 16 kHz. Then speech envelope was extracted using a pass-band filter between 2.25 and 5 Hz. The phase was computed using the Hilbert transform. To quantify the degree of coordination of the verbal interaction (VCI) we computed the phase locking value using mne_connectivity python function spectral_connectivity_time with the method 'plv' based on the phase locking value index proposed by *Lachaux et al., 1999*. Phase locking was computed for each trial on speech temporal envelope (*Figure 1B*), resulting in 24 or 48 VCIs per patient depending on the number of sessions performed. To assess the effect of the coupling parameter, we computed a linear mixed-model contrasting *in-phase coupling trials and trials with a coupling set towards a 180° shift* (lmer(VCI ~ K + (1|participant))). As expected, because of the varying adaptive behaviour of the VP, VCI varies across trials, indexing more or less efficient coordinative behaviour (*Figure 1C*). Moreover, in order to estimate whether the level of performance was greater than chance level, we computed, for each patient and each trial, a null distribution obtained by randomly shifting the phase between the VP and the patient speech (500 times, see *Figure 2—figure supplement 1*).

### sEEG signal

General preprocessing related to electrodes localisation

To increase spatial sensitivity and reduce passive volume conduction from neighbouring regions (*Mercier et al., 2017*), the signal was offline re-referenced using bipolar montage. That is, for a pair of adjacent electrode contacts, the referencing led to a virtual channel located at the midpoint locations of the original contacts. To precisely localise the channels, a procedure similar to the one used in the iELVis toolbox was applied (*Groppe et al., 2017*). Firstly, we manually identified the location of each channel centroid on the post-implant CT scan using the Gardel software (*Medina Villalon et al., 2018*). Secondly, we performed volumetric segmentation and cortical reconstruction on the pre-implant MRI with the Freesurfer image analysis suite (documented and freely available for download online http://surfer.nmr.mgh.harvard.edu/). This segmentation of the pre-implant MRI with SPM12 provides us with both the tissue probability maps (i.e. grey, white, and cerebrospinal fluid [CSF] probabilities) and the indexed-binary representations (i.e. either grey, white, CSF, bone, or soft tissues). This information allowed us to reject electrodes not located in the brain. Thirdly, the post-implant CT scan was coregistered to the pre-implant MRI via a rigid affine transformation and the pre-implant MRI was registered to the MNI template (MNI 152 Linear), via a linear and a non-linear transformation from SPM12 methods (*Penny et al., 2011*), through the FieldTrip toolbox (*Oostenveld et al., 2011*). Fourth, applying the corresponding transformations, we mapped channel locations to the pre-implant MRI brain that was labelled using the volume-based Human Brainnetome Atlas (*Fan et al., 2016*).

## Signal preprocessing

Continuous signal was filtered using (1) a notch filter at 50 Hz and harmonics up to 300 Hz to remove power line artifacts and (2) a bandpass filter between 0.5 and 300 Hz.

To define artefacted channel we used the broadband (raw) signal delimited on the experimental task recording. Channels with a variance greater than 2*IQR (interquartile range, i.e. a non-parametric estimate of the standard deviation) were tagged as artefacted channels (6% of the channels). Channels defined as artefacted were excluded from subsequent analysis.

The continuous signal during the task was then epoched from −0.2 to 14 s relative to the onset of the first stimulus (repetition) of each trial. Generating 24 or 48 epochs depending on the number of sessions each patient had completed. A baseline correction was applied from −0.1 to 0 s. The first 500 ms were discarded from the epoched data to avoid the activity burst generated at the onset of the stimulus.

The continuous signal from the resting state session was also epoched in seventeen 14-s non-overlapping epochs.

## Power spectral density

The power spectral density (PSD) computation was conducted for each channel, with the MNE-python function Epochs.compute_psd, trial-by-trial (epochs) in a range of 125 frequencies, logarithmically scaled, ranging from 0.5 to 125 Hz. Six canonical frequency bands were investigated by averaging the power spectra between their respective boundaries: delta (1–4 Hz), theta (4–8 Hz), alpha (8–13 Hz), beta (13–30 Hz), low gamma (30–50 Hz), and HFa (70–125 Hz).

## Global effect

The global effect of the task (vs period of rest) was computed on each frequency band by subtracting first the mean resting state activity from the mean experimental activity and then dividing by the mean resting state activity. This approach has the advantage of centring the magnitude before expressing it in percentage (*Mercier et al., 2022*). For each frequency band and channel, the statistical difference between task activity and the baseline (resting) was estimated with permutation tests ($N = 1000$) using the SciPy library.

## Coupling behavioural and neurophysiological data

Behavioural speech and neurophysiological data were jointly analysed using three approaches. In the first approach, a two-step procedure was used. Firstly, for each frequency band, channel and trial, we computed the mean power. Then, for each frequency band and channel, we computed the correlation across trials using a non-parametric correlation metric (Spearman) between the power and the VCI (between VP and patient speech). A rho value was thus attributed to each channel and frequency band. The significance was assessed using a permutation approach, similar to the one used for the global effect (see above). In the second approach, we computed the PAC between speech phase and HFa. More precisely, we used two measures for the phase: (1) the phase of the speech envelope of the VP, corresponding to the speech input, (2) the instantaneous phase difference between VP and patient phases, corresponding to the instantaneous coordination of participant and VP (see *Figure 2B*). As for power, we used the HFa from 70 to 125 Hz (considered as a proxy for activity population-level spiking activity of neurons; *Buzsáki et al., 2012*). The computation was performed using Tensorpac, an open-source Python toolbox for tensor-based PAC measurement in electrophysiological brain signals (*Combrisson et al., 2020*). In the third approach, we assessed whether the phase-amplitude relationship (or coupling) depends upon the anticipatory (negative delays) or compensatory (positive delays) behaviour between the VO and the patients' speech. We computed the average delay in each trial using a cross-correlation approach on speech signals (between patient and VP) with the MATLAB function *xcorr*. A median split (patient-specific; average median split = 0 ms, average sd = 24 ms) was applied to conserve a sufficient amount of data, classifying trials below the median as 'anticipatory behaviour' and trials above the median as 'compensatory behaviour'. Then we conducted the PAC analyses on positive and negative trials separately.

## Clustering analysis

A spatial unsupervised clustering analysis (*k*-means) was conducted on all significant channels, separately for the global effect (task vs rest) and brain–behaviour correlation analyses. Precisely, we used the silhouette score method on the *k*-means result (*Rousseeuw, 1987*; *Shahapure and Nicholas, 2020*). This provides a measure of consistency within clusters of data (or alternatively the goodness of clusters separation). Scores were computed for different numbers of clusters (from 2 to 10). The highest silhouette score indicates the optimal number of clusters. Clustering and silhouette scores were computed using the Scikit-learn's Kmeans and silhouette score function (*Pedregosa et al., 2011*). The statistical difference between spatial clustering in global effect and brain–behaviour correlation was estimated with a linear model using the R function *lm* (stat package), post hoc comparisons were corrected for multiple comparisons using the Tukey test (lsmeans R package; *Lenth, 2016*). The statistical difference between clustering in global effect and behaviour correlation across the number of clusters was estimated using permutation tests ($N = 1000$) by computing the silhouette score difference between the two conditions.

## Acknowledgements

We thank all patients for their willing participation and all the personnel from the epileptology unit.

## Additional information

### Funding

| Funder | Grant reference number | Author |
|---|---|---|
| Agence Nationale de la Recherche | ANR-21-CE28-0010 | Daniele Schön |
| Agence Nationale de la Recherche | ANR-20-CE28-0007-01 | Benjamin Morillon |
| European Research Council | ERC-CoG-101043344 | Benjamin Morillon |
| Fondation Pour l'Audition | FPA RD-2022-09 | Benjamin Morillon |
| NeuroMarseille | ANR-17-EURE-0029 | Benjamin Morillon Daniele Schön |
| Institute of Convergence ILCB | ANR-16-CONV-0002 | Benjamin Morillon Leonardo Lancia |
| Investissements d'Avenir and the Excellence Initiative of Aix Marseille University | AMX-19-IET-004 | Benjamin Morillon Daniele Schön |
| Agence Nationale de la Recherche | ANR21-CE28-0015-01 | Leonardo Lancia |

The funders had no role in study design, data collection, and interpretation, or the decision to submit the work for publication.

### Author contributions

Isaïh Schwab-Mohamed, Conceptualization, Data curation, Formal analysis, Validation, Investigation, Visualization, Methodology, Writing – original draft, Writing – review and editing; Manuel R Mercier, Data curation, Formal analysis, Validation, Visualization, Methodology, Writing – original draft, Writing – review and editing; Agnès Trebuchon, Resources, Data curation, Validation, Visualization, Methodology, Writing – review and editing; Benjamin Morillon, Formal analysis, Funding acquisition, Validation, Visualization, Writing – original draft, Writing – review and editing; Leonardo Lancia, Conceptualization, Resources, Data curation, Software, Formal analysis, Supervision, Validation, Investigation, Visualization, Methodology, Writing – original draft, Project administration, Writing – review and editing, Wrote the code for the experimental setup, Funding acquisition; Daniele Schön, Conceptualization, Resources, Data curation, Software, Formal analysis, Supervision, Funding acquisition, Validation, Investigation, Visualization, Methodology, Writing – original draft, Project administration, Writing – review and editing

### Author ORCIDs

Isaïh Schwab-Mohamed (i) https://orcid.org/0009-0005-9105-7145
Manuel R Mercier (i) https://orcid.org/0000-0001-6358-4734
Agnès Trebuchon (i) https://orcid.org/0000-0002-8632-3454
Benjamin Morillon (i) https://orcid.org/0000-0002-0049-064X
Leonardo Lancia (i) https://orcid.org/0009-0005-3805-4201
Daniele Schön (i) https://orcid.org/0000-0003-4472-4150

### Ethics

Patients provided written informed consent prior to the experimental session and the experimental protocol was approved by the Institutional Review Board of the French Institute of Health (IRB00003888).

Reviewer #1 (Public review): https://doi.org/10.7554/eLife.99547.4.sa1
Reviewer #2 (Public review): https://doi.org/10.7554/eLife.99547.4.sa2
Author response https://doi.org/10.7554/eLife.99547.4.sa3

## Additional files

### Supplementary files
MDAR checklist

### Data availability
The raw data investigated in the current manuscript is privileged patient data. Because of this, the conditions of our ethics approval do not permit public archiving of anonymised study data. Readers seeking access to the data should contact Dr. Daniele Schön (daniele.schon@univ-amu.fr). Access will be granted to named individuals in accordance with ethical procedures governing the reuse of clinical data, including completion of a formal data sharing agreement. Data analyses were performed using custom scripts in Python, Matlab, and R, which are available on GitHub along with preprocessed data necessary to reproduce the figures and results (https://github.com/isaiih/VerbalCoord-NeuralDyna, copy archived at *Mohamed, 2025*). Readers seeking more details about the Virtual Partner (VP) model should contact Leonardo Lancia (leonardo.lancia@cnrs.fr).

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
