## [Editor Report · eLife Assessment]

This paper reports on an **important** study that aims to move beyond current experimental approaches in speech production by (1) investigating speech in the context of a fully interactive task and (2) employing advanced methodology to record intracranial brain activity. Together these allow for examination of the unfolding temporal dynamics of brain-behaviour relationships during interactive speech. This approach and the analyses presented in support of the authors' claims pose **convincing** evidence.

---

## [Referee Report · Reviewer #1 (Public review)]

Summary:

This paper reports an intracranial SEEG study of speech coordination, where participants synchronize their speech output with a virtual partner that is designed to vary its synchronization behavior. This allows the authors to identify electrodes throughout the left hemisphere of the brain that have activity (both power and phase) that correlates with the degree of synchronization behavior. They find that high-frequency activity in secondary auditory cortex (superior temporal gyrus) is correlated to synchronization, in contrast to primary auditory regions. Furthermore, activity in inferior frontal gyrus shows a significant phase-amplitude coupling relationship that is interpreted as compensation for deviation from synchronized behavior with the virtual partner.

Strengths:

(1) The development of a virtual partner model trained for each individual participant, which can dynamically vary its synchronization to the participant's behavior in real time, is novel and exciting.

(2) Understanding real-time temporal coordination for behaviors like speech is a critical and understudied area.

(3) The use of SEEG provides the spatial and temporal resolution necessary to address the complex dynamics associated with the behavior.

(4) The paper provides some results that suggest a role for regions like IFG and STG in the dynamic temporal coordination of behavior both within an individual speaker and across speakers performing a coordination task.

---

## [Referee Report · Reviewer #2 (Public review)]

Summary:

This paper investigates the neural underpinnings of an interactive speech task requiring verbal coordination with another speaker. To achieve this, the authors recorded intracranial brain activity from the left (and to a lesser extent, the right) hemisphere in a group of drug-resistant epilepsy patients while they synchronised their speech with a 'virtual partner'. Crucially, the authors were able to manipulate the degree of success of this synchronisation by programming the virtual partner to either actively synchronise or desynchronise their speech with the participant, or else to not vary its speech in response to the participant (making the synchronisation task purely one-way). Using such a paradigm, the authors identified different brain regions that were either more sensitive to the speech of the virtual partner (primary auditory cortex), or more sensitive to the degree of verbal coordination (i.e. synchronisation success) with the virtual partner (left secondary auditory cortex and bilateral IFG). Such sensitivity was measured by (1) calculating the correlation between the index of verbal coordination and mean power within a range of frequency bands across trials, and (2) calculating the phase-amplitude coupling between the behavioural and brain signals within single trials (using the power of high-frequency neural activity only). Overall, the findings help to elucidate some of the brain areas involved in interactive speaking behaviours, particularly highlighting high-frequency activity of the bilateral IFG as a potential candidate supporting verbal coordination.

Strengths:

This study provides the field with a convincing demonstration of how to investigate speaking behaviours in more complex situations that share many features with real-world speaking contexts e.g. simultaneous engagement of speech perception and production processes, the presence of an interlocutor and the need for inter-speaker coordination. The findings thus go beyond previous work that has typically studied solo speech production in isolation, and represent a significant advance in our understanding of speech as a social and communicative behaviour. It is further an impressive feat to develop a paradigm in which the degree of cooperativity of the synchronisation partner can be so tightly controlled; in this way, this study combines the benefits of using pre-recorded stimuli (namely, the high degree of experimental control) with the benefits of using a live synchronisation partner (allowing the task to be truly two-way interactive, an important criticism of other work using pre-recorded stimuli). A further key strength of the study lies in its employment of stereotactic EEG to measure brain responses with both high temporal and spatial resolution, an ideal method for studying the unfolding relationship between neural processing and this dynamic coordination behaviour.

Weaknesses:

One limitation of the current study is the relatively sparse coverage of the right hemisphere by the implanted electrodes (91 electrodes in the right compared to 145 in the left). Of course, electrode location is solely clinically motivated, and so the authors did not have control over this. In a previous version of this article, the authors therefore chose not to include data from the right hemisphere in reported analyses. However, after highlighting previous literature suggesting that the right hemisphere likely has high relevance to verbal coordination behaviours such as those under investigation here, the authors have now added analyses of the right hemisphere data to the results. These confirm an involvement of the right hemisphere in this task, largely replicating left hemisphere results. Some hemispheric differences were found in responses within the STG; however, interpretation should be tempered by an awareness of the relatively sparse coverage of the right hemisphere meaning that some regions have very few electrodes, resulting in reduced statistical power.

---

## [Author Response]

The following is the authors’ response to the previous reviews

**Recommendations for the authors:**

**Reviewer #1 (Recommendations for the authors):**
(1) The use of the term "language network" throughout is unclear. Does this refer to work by Ev Fedorenko (i.e., does it distinguish language from other cognitive and sensorimotor domains)? There does not seem to be much in the behavior presented here that aligns with an interpretation about language per se.

We understand the reviewer’s point according to the work by Evelina Fedorenko considering this distinction. It is important to precise that in our present study we did not refer to her work when using the term “language network”.

(2) Fig 4A: the "B" is missing on the figure panel to denote which Broadmann areas are shown.

We updated the figure panel by adding the “B” for more clarity.

**Reviewer #2 (Recommendations for the authors):**
I think it would be worth mentioning the relatively sparse coverage of the right hemisphere in your abstract.

We agree with this suggestion, we updated the abstract as follows :

“Our use of language, which is profoundly social in nature, essentially takes place in interactive contexts and is shaped by precise coordination dynamics that interlocutors must observe. Thus, language interaction is highly demanding on fast adjustment of speech production. Here, we developed a real-time coupled-oscillators virtual partner that allows - by changing the coupling strength parameters - to modulate the ability to synchronise speech with a virtual speaker. Then, we recorded the intracranial brain activity of 16 patients with drug-resistant epilepsy while they performed a verbal coordination task with the virtual partner (VP). More precisely, patients had to repeat short sentences synchronously with the VP. This synchronous speech task is efficient to highlight both the dorsal and ventral language pathways. Importantly, combining time-resolved verbal coordination and neural activity shows more spatially differentiated patterns and different types of neural sensitivity along the dorsal pathway. More precisely, high-frequency activity in left secondary auditory regions is highly sensitive to verbal coordinative dynamics, while primary regions are not. Finally, while bilateral engagement was observed in the high-frequency activity of the IFG BA44— which seems to index online coordinative adjustments that are continuously required to compensate deviation from synchronisation—interpretation of right hemisphere involvement should be approached cautiously due to relatively sparse electrode coverage. These findings illustrate the possibility and value of using a fully dynamic, adaptive and interactive language task to gather deeper understanding of the subtending neural dynamics involved in speech perception, production as well as their interaction.”

There are a few places in your results section which haven't been updated to reflect the fact that some sections refer only to the left hemisphere e.g.Page 11 line 347: "Overall, neural responses are present in all six canonical frequency bands" I think this should be "In the left hemisphere, neural responses are present...".Page 12 line 355: "As expected, the whole language network is strongly involved..." I think this should be "As expected, the whole left hemisphere language network is strongly involved". Page 17 (third paragraph of the discussion): "The observed negative correlation between verbal coordination and high-frequency activity (HFa) in STG BA22" I think this should be "in left STG BA22".

We thank the reviewer for highlighting these important points. The updated lines are as follows:

Page 11 line 348: ”In the left hemisphere, neural responses are present in all six canonical frequency bands…”

Page 12 line 356: ”As expected, the whole left hemisphere language network is strongly involved..." Page 17 lines 502-503 : “The observed negative correlation between verbal coordination and highfrequency activity (HFa) in left STG BA22 suggests a suppression of neural responses as the degree of behavioural synchrony increases.”